# Large Scale computation of Means and Clusters for Persistence Diagrams using Optimal Transport

**Théo Lacombe**
Datashape
Inria Saclay
theo.lacombe@inria.fr

**Marco Cuturi**
Google Brain, and
CREST, ENSAE
cuturi@google.com

**Steve Oudot**
Datashape
Inria Saclay
steve.oudot@inria.fr

## Abstract

Persistence diagrams (PDs) are now routinely used to summarize the underlying topology of complex data. Despite several appealing properties, incorporating PDs in learning pipelines can be challenging because their natural geometry is not Hilbertian. Indeed, this was recently exemplified in a string of papers which show that the simple task of averaging a few PDs can be computationally prohibitive. We propose in this article a tractable framework to carry out standard tasks on PDs at scale, notably evaluating distances, estimating barycenters and performing clustering. This framework builds upon a reformulation of PD metrics as optimal transport (OT) problems. Doing so, we can exploit recent computational advances: the OT problem on a planar grid, when regularized with entropy, is convex can be solved in linear time using the Sinkhorn algorithm and convolutions. This results in scalable computations that can stream on GPUs. We demonstrate the efficiency of our approach by carrying out clustering with diagrams metrics on several thousands of PDs, a scale never seen before in the literature.

## 1  Introduction

Topological data analysis (TDA) has been used successfully in a wide array of applications, for instance in medical (Nicolau et al., 2011) or material (Hiraoka et al., 2016) sciences, computer vision (Li et al., 2014) or to classify NBA players (Lum et al., 2013). The goal of TDA is to exploit and account for the complex topology (connectivity, loops, holes, etc.) seen in modern data. The tools developed in TDA are built upon persistent homology theory (Edelsbrunner et al., 2000; Zomorodian & Carlsson, 2005; Edelsbrunner & Harer, 2010) whose main output is a descriptor called a *persistence diagram* (PD) which encodes in a compact form—roughly speaking, a point cloud in the upper triangle of the square $[0,1]^2$—the topology of a given space or object at all scales.

**Statistics on PDs.** Persistence diagrams have appealing properties: in particular they have been shown to be stable with respect to perturbations of the input data (Cohen-Steiner et al., 2007; Chazal et al., 2009, 2014). This stability is measured either in the so called *bottleneck* metric or in the $p$-th diagram distance, which are both distances that compute optimal partial matchings. While theoretically motivated and intuitive, these metrics are by definition very costly to compute. Furthermore, these metrics are not Hilbertian, preventing a faithful application of a large class of standard machine learning tools ($k$-means, PCA, SVM) on PDs.

**Related work.** To circumvent the non-Hilbertian nature of the space of PDs, one can of course map diagrams onto simple feature vectors. Such features can be either finite dimensional (Carrière et al., 2015; Adams et al., 2017), or infinite through kernel functions (Reininghaus et al., 2015; Bubenik, 2015; Carrière et al., 2017). A known drawback of kernel approaches on a rich geometric space such as that formed by PDs is that once PDs are mapped as feature vectors, any ensuing analysis remains in the space of such features (the "inverse image" problem inherent to kernelization). They are therefore

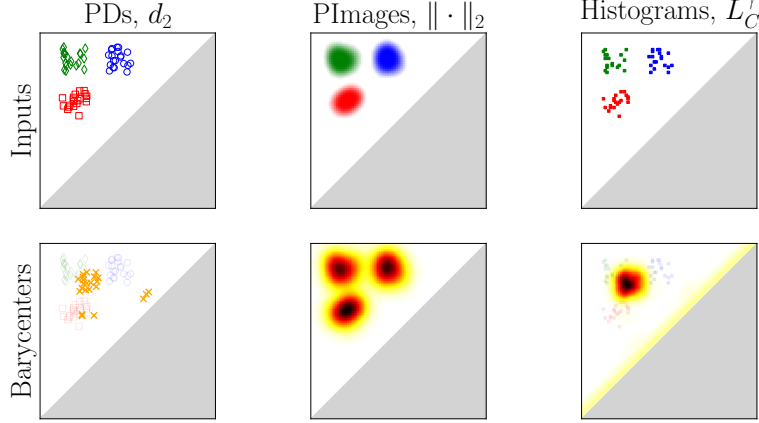

Figure 1: Illustration of differences between Fréchet means with Wasserstein and Euclidean geometry. The top row represents input data, namely persistence diagrams (left), vectorization of PDs as persistence images in $\mathbb{R}^{100 \times 100}$ (middle, (Adams et al., 2017)), and discretization of PDs as histograms (right). The bottom row represents the estimated barycenters (orange scale) with input data (shaded), using the approach of Turner et al. (2014) (left), the arithmetic mean of persistence images (middle) and our optimal tranport based approach (right).

not helpful to carry out simple tasks in the space of PDs, such as that of averaging PDs, namely computing the Fréchet mean of a family of PDs. Such problems call for algorithms that are able to optimize directly in the space of PDs, and were first addressed by Mileyko et al. (2011) and Turner (2013). Turner et al. (2014) provides an algorithm that converges to a local minimum of the Fréchet function by successive iterations of the Hungarian algorithm. However, the Hungarian algorithm does not scale well with the size of diagrams, and non-convexity yields potentially convergence to bad local minima.

**Contributions.** We reformulate the computation of diagram metrics as an optimal transport (OT) problem, opening several perspectives, among them the ability to benefit from entropic regularization (Cuturi, 2013). We provide a new numerical scheme to bound OT metrics, and therefore diagram metrics, with additive guarantees. Unlike previous approximations of diagram metrics, ours can be parallelized and implemented efficiently on GPUs. These approximations are also differentiable, leading to a scalable method to compute barycenters of persistence diagrams. In exchange for a discretized approximation of PDs, we recover a convex problem, unlike previous formulations of the barycenter problem for PDs. We demonstrate the scalability of these two advances (accurate approximation of the diagram metric at scale and barycenter computation) by providing the first tractable implementation of the $k$-means algorithm in the space of PDs.

**Notations for matrix and vector manipulations.** When applied to matrices or vectors, operators $\exp, \log$, division, are always meant *element-wise*. $u \odot v$ denotes element-wise multiplication (Hadamard product) while $Ku$ denotes the matrix-vector product of $K \in \mathbb{R}^{d \times d}$ and $u \in \mathbb{R}^d$.

## 2 Background on OT and TDA

**OT.** Optimal transport is now widely seen as a central tool to compare probability measures (Villani, 2003, 2008; Santambrogio, 2015). Given a space $\mathcal{X}$ endowed with a cost function $c : \mathcal{X} \times \mathcal{X} \to \mathbb{R}_+$, we consider two *discrete* measures $\mu$ and $\nu$ on $\mathcal{X}$, namely measures that can be written as positive combinations of diracs, $\mu = \sum_{i=1}^n a_i \delta_{x_i}, \nu = \sum_{j=1}^m b_j \delta_{y_j}$ with weight vectors $a \in \mathbb{R}_+^n, b \in \mathbb{R}_+^m$ satisfying $\sum_i a_i = \sum_j b_j$ and all $x_i, y_j$ in $\mathcal{X}$. The $n \times m$ cost matrix $C = (c(x_i, y_j))_{ij}$ and the transportation polytope $\Pi(a, b) := \{P \in \mathbb{R}_+^{n \times m} | P\mathbf{1}_m = a, P^T\mathbf{1}_n = b\}$ define an optimal transport problem whose optimum $\mathbf{L}_C$ can be computed using either of two linear programs, dual to each other,

$$\mathbf{L}_C(\mu, \nu) := \min_{P \in \Pi(a,b)} \langle P, C \rangle = \max_{(\alpha, \beta) \in \Psi_C} \langle \alpha, a \rangle + \langle \beta, b \rangle \tag{1}$$

where $\langle \cdot, \cdot \rangle$ is the Frobenius dot product and $\Psi_C$ is the set of pairs of vectors $(\alpha, \beta)$ in $\mathbb{R}^n \times \mathbb{R}^m$ such that their tensor sum $\alpha \oplus \beta$ is smaller than $C$, namely $\forall i, j, \alpha_i + \beta_j \leq C_{ij}$. Note that when $n = m$ and all weights $a$ and $b$ are uniform and equal, the problem above reduces to the computation of an *optimal matching*, that is a permutation $\sigma \in \mathfrak{S}_n$ (with a resulting optimal plan $P$ taking the form $P_{ij} = 1_{\sigma(i)=j}$). That problem has clear connections with diagram distances, as shown in §3.

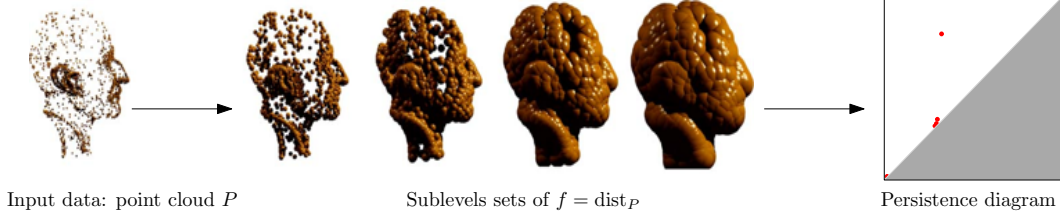

Input data: point cloud $P$          Sublevels sets of $f = \mathrm{dist}_P$          Persistence diagram

Figure 2: Sketch of persistent homology. $\mathbb{X} = \mathbb{R}^3$ and $f(x) = \min_{p \in P} \|x - p\|$ so that sublevel sets of $f$ are unions of balls centered at the points of $P$. First (resp second) coordinate of points in the persistence diagram encodes appearance scale (resp disappearance) of cavities in the sublevel sets of $f$. The isolated red point accounts for the presence of a *persistent* hole in the sublevel sets, inferring the underlying spherical geometry of the input point cloud.

**Entropic Regularization.** Solving the optimal transport problem is intractable for large data. Cuturi proposes to consider a regularized formulation of that problem using entropy, namely:

$$\mathbf{L}_C^\gamma(a, b) := \min_{P \in \Pi(a,b)} \langle P, C \rangle - \gamma h(P) \tag{2}$$

$$= \max_{\alpha \in \mathbb{R}^n, \beta \in \mathbb{R}^m} \langle \alpha, a \rangle + \langle \beta, b \rangle - \gamma \sum_{i,j} e^{\frac{\alpha_i + \beta_j - C_{i,j}}{\gamma}}, \tag{3}$$

where $\gamma > 0$ and $h(P) := -\sum_{ij} P_{ij}(\log P_{ij} - 1)$. Because the negentropy is 1-strongly convex, that problem has a unique solution $P^\gamma$ which takes the form, using first order conditions,

$$P^\gamma = \mathrm{diag}(u^\gamma) K \mathrm{diag}(v^\gamma) \in \mathbb{R}^{n \times m}, \tag{4}$$

where $K = e^{-\frac{C}{\gamma}}$ (term-wise exponentiation), and $(u^\gamma, v^\gamma) \in \mathbb{R}^n \times \mathbb{R}^m$ is a fixed point of the Sinkhorn map (term-wise divisions):

$$S : (u, v) \mapsto \left( \frac{a}{Kv}, \frac{b}{K^T u} \right). \tag{5}$$

Note that this fixed point is the limit of any sequence $(u_{t+1}, v_{t+1}) = S(u_t, v_t)$, yielding a straightforward algorithm to estimate $P^\gamma$. Cuturi considers the transport cost of the optimal regularized plan, $\mathbf{S}_C^\gamma(a, b) := \langle P^\gamma, C \rangle = (u^\gamma)^T (K \odot C) v^\gamma$, to define a *Sinkhorn divergence* between $a, b$ (here $\odot$ is the term-wise multiplication). One has that $\mathbf{S}_C^\gamma(a, b) \to \mathbf{L}_C(a, b)$ as $\gamma \to 0$, and more precisely $P^\gamma$ converges to the optimal transport plan solution of (1) with maximal entropy. That approximation can be readily applied to any problem that involves terms in $\mathbf{L}_C$, notably barycenters (Cuturi & Doucet, 2014; Solomon et al., 2015; Benamou et al., 2015).

**Eulerian setting.** When the set $\mathcal{X}$ is finite with cardinality $d$, $\mu$ and $\nu$ are entirely characterized by their probability weights $a, b \in \mathbb{R}_+^d$ and are often called *histograms* in a *Eulerian* setting. When $\mathcal{X}$ is not discrete, as when considering the plane $[0, 1]^2$, we therefore have a choice of representing measures as sums of diracs, encoding their information through locations, or as discretized histograms on a planar grid of arbitrary granularity. Because the latter setting is more effective for entropic regularization (Solomon et al., 2015), this is the approach we will favor in our computations.

**Persistent homology and Persistence Diagrams.** Given a topological space $\mathbb{X}$ and a real-valued function $f : \mathbb{X} \to \mathbb{R}$, persistent homology provides—under mild assumptions on $f$, taken for granted in the remaining of this article—a topological signature of $f$ built on its *sublevel sets* $\left( f^{-1}((-\infty, t]) \right)_{t \in \mathbb{R}}$, and called a *persistence diagram* (PD), denoted as $\mathrm{Dgm}(f)$. In practice, it is of the form $\mathrm{Dgm}(f) = \sum_{i=1}^n \delta_{x_i}$, namely a point measure with finite support included in $\mathbb{R}_>^2 := \{(s, t) \in \mathbb{R}^2 | s < t\}$. Each point $(s, t)$ in $\mathrm{Dgm}(f)$ can be understood as a topological feature (connected component, loop, hole...) which appears at scale $s$ and disappears at scale $t$ in the sublevel sets of $f$. Comparing the persistence diagrams of two functions $f, g$ measures their difference from a topological perspective: presence of some topological features, difference in appearance scales, etc. The space of PDs is naturally endowed with a partial matching metric defined as ($p \geq 1$):

$$d_p(D_1, D_2) := \left( \min_{\zeta \in \Gamma(D_1, D_2)} \sum_{(x,y) \in \zeta} \|x - y\|_p^p + \sum_{s \in D_1 \cup D_2 \setminus \zeta} \|s - \pi_\Delta(s)\|_p^p \right)^{\frac{1}{p}}, \tag{6}$$

where $\Gamma(D_1, D_2)$ is the set of all partial matchings between points in $D_1$ and points in $D_2$ and $\pi_\Delta(s)$ denotes the orthogonal projection of an (unmatched) point $s$ to the diagonal $\{(x, x) \in \mathbb{R}^2, x \in \mathbb{R}\}$. The mathematics of OT and diagram distances share a key idea, that of matching, but differ on an important aspect: diagram metrics can cope, using the diagonal as a sink, with measures that have a varying total number of points. We solve this gap by leveraging an unbalanced formulation for OT.

## 3   Fast estimation of diagram metrics using Optimal Transport

In the following, we start by explicitly formulating (6) as an optimal transport problem. Entropic smoothing provides us a way to approximate (6) with controllable error. In order to benefit mostly from that regularization (matrix parallel execution, convolution, GPU—as showcased in (Solomon et al., 2015)), implementation requires specific attention, as described in propositions 2, 3, 4.

**PD metrics as Optimal Transport.**   The main differences between (6) and (1) are that PDs do not generally have the same mass, i.e. number of points (counted with multiplicity), and that the diagonal plays a special role by allowing to match any point $x$ in a given diagram with its orthogonal projection $\pi_\Delta(x)$ onto the diagonal. Guittet's formulation for partial transport (2002) can be used to account for this by creating a "sink" bin corresponding to that diagonal and allowing for different total masses. The idea of representing the diagonal as a single point already appears in the bipartite graph problem of Edelsbrunner & Harer (2010) (Ch.VIII). The important aspect of the following proposition is the clarification of the partial matching problem (6) as a standard OT problem (1).

Let $\mathbb{R}^2_> \cup \{\Delta\}$ be $\mathbb{R}^2_>$ extended with a *unique* virtual point $\{\Delta\}$ encoding the diagonal. We introduce the linear operator $\mathbf{R}$ which, to a finite non-negative measure $\mu$ supported on $\mathbb{R}^2_>$, associates a dirac on $\Delta$ with mass equal to the total mass of $\mu$, namely $\mathbf{R} : \mu \mapsto |\mu|\delta_\Delta$.

**Proposition 1.** *Let $D_1 = \sum_{i=1}^{n_1} \delta_{x_i}$ and $D_2 = \sum_{j=1}^{n_2} \delta_{y_j}$ be two persistence diagrams with respectively $n_1$ points $x_1 \ldots x_{n_1}$ and $n_2$ points $y_1 \ldots y_{n_2}$. Let $p \geq 1$. Then:*

$$d_p(D_1, D_2)^p = \mathbf{L}_C(D_1 + \mathbf{R}D_2, D_2 + \mathbf{R}D_1), \tag{7}$$

*where $C$ is the cost matrix with block structure*

$$C = \begin{pmatrix} \widehat{C} & u \\ v^T & 0 \end{pmatrix} \in \mathbb{R}^{(n_1+1) \times (n_2+1)}, \tag{8}$$

*where $u_i = \|x_i - \pi_\Delta(x_i)\|_p^p, v_j = \|y_j - \pi_\Delta(y_j)\|_p^p, \widehat{C}_{ij} = \|x_i - y_j\|_p^p, for \ i \leq n_1, j \leq n_2$.*

The proof seamlessly relies on the fact that, when transporting point measures with the *same* mass (number of points counted with multiplicity), the optimal transport problem is equivalent to an optimal matching problem (see §2). Details are left to the supplementary material.

**Entropic approximation of diagram distances.**   Following the correspondance established in Proposition 1, entropic regularization can be used to approximate the diagram distance $d_p(\cdot, \cdot)$. Given two persistence diagrams $D_1, D_2$ with respective masses $n_1$ and $n_2$, let $n := n_1 + n_2$, $a = (\mathbf{1}_{n_1}, n_2) \in \mathbb{R}^{n_1+1}, b = (\mathbf{1}_{n_2}, n_1) \in \mathbb{R}^{n_2+1}$, and $P_t^\gamma = \mathrm{diag}(u_t^\gamma)K\mathrm{diag}(v_t^\gamma)$ where $(u_t^\gamma, v_t^\gamma)$ is the output after $t$ iterations of the Sinkhorn map (5). Adapting the bounds provided by Altschuler et al. (2017), we can bound the error of approximating $d_p(D_1, D_2)^p$ by $\langle P_t^\gamma, C \rangle$:

$$|d_p(D_1, D_2)^p - \langle P_t^\gamma, C \rangle| \leq 2\gamma n \log(n) + \mathrm{dist}(P_t^\gamma, \Pi(a, b))\|C\|_\infty \tag{9}$$

where $\mathrm{dist}(P, \Pi(a, b)) := \|P\mathbf{1} - a\|_1 + \|P^T\mathbf{1} - b\|_1$ (that is, error on marginals).

Dvurechensky et al. (2018) prove that iterating the Sinkhorn map (5) gives a plan $P_t^\gamma$ satisfying $\mathrm{dist}(P_t^\gamma, \Pi(a, b)) < \varepsilon$ in $\mathcal{O}\left(\frac{\|C\|_\infty^2}{\gamma\varepsilon} + \ln(n)\right)$ iterations. Given (9), a natural choice is thus to take $\gamma = \frac{\varepsilon}{n\ln(n)}$ for a desired precision $\varepsilon$, which lead to a total of $\mathcal{O}\left(\frac{n\ln(n)\|C\|_\infty^2}{\varepsilon^2}\right)$ iterations in the Sinkhorn loop. These results can be used to pre-tune parameters $t$ and $\gamma$ to control the approximation error due to smoothing. However, these are worst-case bounds, controlled by max-norms, and are often too pessimistic in practice. To overcome this phenomenon, we propose on-the-fly error control, using approximate solutions to the smoothed primal (2) and dual (3) optimal transport problems, which provide upper and lower bounds on the optimal transport cost.

**Upper and Lower Bounds.**   The Sinkhorn algorithm, after at least one iteration ($t \geq 1$), produces feasible *dual* variables $(\alpha_t^\gamma, \beta_t^\gamma) = (\gamma \log(u_t^\gamma), \gamma \log(v_t^\gamma)) \in \Psi_C$ (see below (1) for a definition).

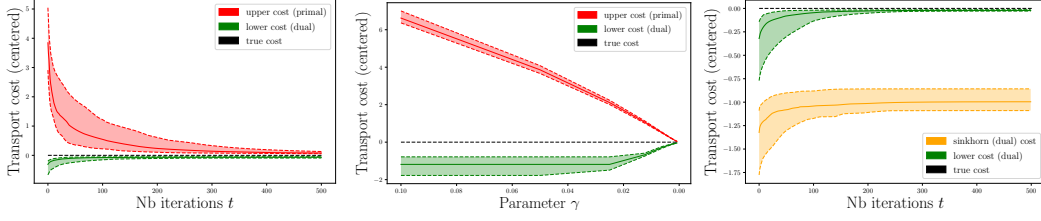

Figure 3: *(Left)* $M_t^\gamma := \langle R_t^\gamma, C \rangle$ (red) and $m_t^\gamma := \langle \alpha_t^{c\bar{c}}, a \rangle + \langle \alpha_t^c, b \rangle$ (green) as a function of $t$, the number of iterations of the Sinkhorn map ($t$ ranges from 1 to 500, with fixed $\gamma = 10^{-3}$). *(Middle)* Final $M^\gamma$ (red) and $m^\gamma$ (green) provided by Alg.1, computed for decreasing $\gamma$s, ranging from $10^{-1}$ to $5.10^{-4}$. For each value of $\gamma$, Sinkhorn loop is run until $d(P_t^\gamma, \Pi(a,b)) < 10^{-3}$. Note that the $\gamma$-axis is flipped. *(Right)* Influence of $c\bar{c}$-transform for the Sinkhorn dual cost. *(orange)* The dual cost $\langle \alpha_t^\gamma, a \rangle + \langle \beta_t^\gamma, b \rangle$, where $(\alpha_t^\gamma, \beta_t^\gamma)$ are Sinkhorn dual variables (before the $C$-transform). *(green)* Dual cost after $C$-transform, i.e. with $((\alpha_t^\gamma)^{c\bar{c}}, (\alpha_t^\gamma)^c)$. Experiment run with $\gamma = 10^{-3}$ and 500 iterations.

Their objective value, as measured by $\langle \alpha_t^\gamma, a \rangle + \langle \beta_t^\gamma, b \rangle$, performs poorly as a lower bound of the true optimal transport cost (see Fig. 3 and §5 below) in most of our experiments. To improve on this, we compute the so called *C-transform* $(\alpha_t^\gamma)^c$ of $\alpha_t^\gamma$ (Santambrogio, 2015, §1.6), defined as:

$$\forall j, (\alpha_t^\gamma)_j^c = \max_i \{C_{ij} - \alpha_i\}, j \leq n_2 + 1.$$

Applying a $C^T$-transform on $(\alpha_t^\gamma)^c$, we recover two vectors $(\alpha_t^\gamma)^{c\bar{c}} \in \mathbb{R}^{n_1+1}, (\alpha_t^\gamma)^c \in \mathbb{R}^{n_2+1}$. One can show that for any feasible $\alpha, \beta$, we have that (Peyré & Cuturi, 2018, Prop 3.1)

$$\langle \alpha, a \rangle + \langle \beta, b \rangle \leq \langle \alpha^{c\bar{c}}, a \rangle + \langle \alpha^c, b \rangle .$$

When $C$'s top-left block is the squared Euclidean metric, this problem can be cast as that of computing the *Moreau envelope* of $\alpha$. In a Eulerian setting and when $\mathcal{X}$ is a finite regular grid which we will consider, we can use either the linear-time Legendre transform or the Parabolic Envelope algorithm (Lucet, 2010, §2.2.1,§2.2.2) to compute the $C$-transform in linear time with respect to the grid resolution $d$.

Unlike dual iterations, the *primal* iterate $P_t^\gamma$ does *not* belong to the transport polytope $\Pi(a,b)$ after a finite number $t$ of iterations. We use the `rounding_to_feasible` algorithm provided by Altschuler et al. (2017) to compute efficiently a feasible approximation $R_t^\gamma$ of $P_t^\gamma$ that does belong to $\Pi(a,b)$. Putting these two elements together, we obtain

$$\underbrace{\langle (\alpha_t^\gamma)^{c\bar{c}}, a \rangle + \langle (\alpha_t^\gamma)^c, b \rangle}_{m_t^\gamma} \leq \mathbf{L}_C(a,b) \leq \underbrace{\langle R_t^\gamma, C \rangle}_{M_t^\gamma}. \tag{10}$$

Therefore, after iterating the Sinkhorn map (5) $t$ times, we have that if $M_t^\gamma - m_t^\gamma$ is below a certain criterion $\varepsilon$, then we can guarantee that $\langle R_t^\gamma, C \rangle$ is *a fortiori* an $\varepsilon$-approximation of $\mathbf{L}_C(a,b)$. Observe that one can also have a relative error control: if one has $M_t^\gamma - m_t^\gamma \leq \varepsilon M_t^\gamma$, then $(1-\varepsilon)M_t^\gamma \leq L_C(a,b) \leq M_t^\gamma$. Note that $m_t^\gamma$ might be negative but can always be replaced by $\max(m_t^\gamma, 0)$ since we know $C$ has non-negative entries (and therefore $\mathbf{L}_C(a,b) \geq 0$), while $M_t^\gamma$ is always non-negative.

**Discretization.** For simplicity, we assume in the remaining that our diagrams have their support in $[0,1]^2 \cap \mathbb{R}_>^2$. From a numerical perspective, encoding persistence diagrams as histograms on the square offers numerous advantages. Given a uniform grid of size $d \times d$ on $[0,1]^2$, we associate to a given diagram $D$ a *matrix-shaped* histogram $\mathbf{a} \in \mathbb{R}^{d \times d}$ such that $\mathbf{a}_{ij}$ is the number of points in $D$ belonging to the cell located at position $(i,j)$ in the grid (we transition to bold-faced small letters to insist on the fact that these histograms must be stored as square matrices). To account for the total mass, we add an extra dimension encoding mass on $\{\Delta\}$. We extend the operator $\mathbf{R}$ to histograms, associating to a histogram $\mathbf{a} \in \mathbb{R}^{d \times d}$ its total mass on the $(d^2+1)$-th coordinate. One can show that the approximation error resulting from that discretization is bounded above by $\frac{1}{d}(|D_1|^{\frac{1}{p}} + |D_2|^{\frac{1}{p}})$ (see the supplementary material).

**Convolutions.** In the Eulerian setting, where diagrams are matrix-shaped histograms of size $d \times d = d^2$, the cost matrix $C$ has size $d^2 \times d^2$. Since we will use large values of $d$ to have low discretization error (typically $d = 100$), instantiating $C$ is usually intractable. However, Solomon et al. (2015)

showed that for regular grids endowed with a separable cost, each Sinkhorn iteration (as well as other key operations such as evaluating Sinkhorn's divergence $\mathbf{S}_C^\gamma$) can be performed using Gaussian convolutions, which amounts to performing matrix multiplications of size $d \times d$, without having to manipulate $d^2 \times d^2$ matrices. Our framework is slightly different due to the extra dimension $\{\Delta\}$, but we show that equivalent computational properties hold. This observation is crucial from a numerical perspective. Our ultimate goal being to efficiently evaluate (11), (12) and (14), we provide implementation details.

Let $(\mathbf{u}, u_\Delta)$ be a pair where $\mathbf{u} \in \mathbb{R}^{d \times d}$ is a matrix-shaped histogram and $u_\Delta \in \mathbb{R}_+$ is a real number accounting for the mass located on the virtual point $\{\Delta\}$. We denote by $\overrightarrow{\mathbf{u}}$ the $d^2 \times 1$ column vector obtained when reshaping $\mathbf{u}$. The $(d^2 + 1) \times (d^2 + 1)$ cost matrix $C$ and corresponding kernel $K$ are given by

$$C = \begin{pmatrix} \widehat{C} & \overrightarrow{\mathbf{c}_\Delta} \\ \overrightarrow{\mathbf{c}_\Delta}^T & 0 \end{pmatrix}, \quad K = \begin{pmatrix} \widehat{K} := e^{-\frac{\widehat{C}}{\gamma}} & \overrightarrow{\mathbf{k}_\Delta} := e^{-\frac{\overrightarrow{\mathbf{c}_\Delta}}{\gamma}} \\ \overrightarrow{\mathbf{k}_\Delta}^T & 1 \end{pmatrix},$$

where $\widehat{C} = (\|(i, i') - (j, j')\|_p^p)_{ii',jj'}$, $\mathbf{c}_\Delta = (\|(i, i') - \pi_\Delta((i, i'))\|_p^p)_{ii'}$. $C$ and $K$ as defined above will never be instantiated, because we can rely instead on $\mathbf{c} \in \mathbb{R}^{d \times d}$ defined as $\mathbf{c}_{ij} = |i - j|^p$ and $\mathbf{k} = e^{-\frac{\mathbf{c}}{\gamma}}$.

**Proposition 2** (Iteration of Sinkhorn map). *The application of $K$ to $(\mathbf{u}, u_\Delta)$ can be performed as:*

$$(\mathbf{u}, u_\Delta) \mapsto \left( \mathbf{k}(\mathbf{k}\mathbf{u}^T)^T + u_\Delta \mathbf{k}_\Delta, \langle \mathbf{u}, \mathbf{k}_\Delta \rangle + u_\Delta \right) \tag{11}$$

*where $\langle \cdot, \cdot \rangle$ denotes the Froebenius dot-product in $\mathbb{R}^{d \times d}$.*

We now introduce $\mathbf{m} := \mathbf{k} \odot \mathbf{c}$ and $\mathbf{m}_\Delta := \mathbf{k}_\Delta \odot \mathbf{c}_\Delta$ ($\odot$ denotes term-wise multiplication).

**Proposition 3** (Computation of $\mathbf{S}_C^\gamma$). *Let $(\mathbf{u}, u_\Delta), (\mathbf{v}, v_\Delta) \in \mathbb{R}^{d \times d + 1}$. The transport cost of $P := \text{diag}(\overrightarrow{\mathbf{u}}, u_\Delta) K \text{diag}(\overrightarrow{\mathbf{v}}, v_\Delta)$ can be computed as:*

$$\langle \text{diag}(\overrightarrow{\mathbf{u}}, u_\Delta) K \text{diag}(\overrightarrow{\mathbf{v}}, v_\Delta), C \rangle = \langle \text{diag}(\overrightarrow{\mathbf{u}}) \widehat{K} \text{diag}(\overrightarrow{\mathbf{v}}), \widehat{C} \rangle + u_\Delta \langle \mathbf{v}, \mathbf{m}_\Delta \rangle + v_\Delta \langle \mathbf{u}, \mathbf{m}_\Delta \rangle, \tag{12}$$

*where the first term can be computed as:*

$$\langle \text{diag}(\overrightarrow{\mathbf{u}}) \widehat{K} \text{diag}(\overrightarrow{\mathbf{v}}), \widehat{C} \rangle = \| \mathbf{u} \odot \left( \mathbf{m}(\mathbf{k}\mathbf{v}^T)^T + \mathbf{k}(\mathbf{m}\mathbf{v}^T)^T \right) \|_1. \tag{13}$$

Finally, consider two histograms $(\mathbf{a}, a_\Delta), (\mathbf{b}, b_\Delta) \in \mathbb{R}^{d \times d} \times \mathbb{R}$, let $R \in \Pi((\mathbf{a}, a_\Delta), (\mathbf{b}, b_\Delta))$ be the rounded matrix of $P$ (see the supplementary material or (Altschuler et al., 2017)). Let $r(P), c(P) \in \mathbb{R}^{d \times d} \times \mathbb{R}$ denote the first and second marginal of $P$ respectively. We introduce (using term-wise min and divisions):

$$X = \min\left( \frac{(\mathbf{a}, a_\Delta)}{r(P)}, \mathbf{1} \right), \quad Y = \min\left( \frac{(\mathbf{b}, b_\Delta)}{c(\text{diag}(X)P)}, \mathbf{1} \right),$$

along with $P' = \text{diag}(X)P\text{diag}(Y)$ and the marginal errors:

$$(e_r, (e_r)_\Delta) = (\mathbf{a}, a_\Delta) - r(P'), \quad (e_c, (e_c)_\Delta) = (\mathbf{b}, b_\Delta) - c(P'),$$

**Proposition 4** (Computation of upper bound $\langle R, C \rangle$). *The transport cost induced by $R$ can be computed as:*

$$\begin{aligned}
\langle R, C \rangle &= \langle \text{diag}(X \odot (\mathbf{u}, u_\Delta)) K \text{diag}(Y \odot (\mathbf{v}, v_\Delta)), C \rangle \\
&+ \frac{1}{\|e_c\|_1 + (e_c)_\Delta} \left( \|e_r^T \mathbf{c} e_c\|_1 + \|e_r \mathbf{c} e_c^T\|_1 + (e_c)_\Delta \langle e_r, \mathbf{c}_\Delta \rangle + (e_r)_\Delta \langle e_c, \mathbf{c}_\Delta \rangle \right).
\end{aligned} \tag{14}$$

*Note that the first term can be computed using* (12)

**Parallelization and GPU.** Using a Eulerian representation is particularly beneficial when applying Sinkhorn's algorithm, as shown by Cuturi (2013). Indeed, the Sinkhorn map (5) only involves matrix-vector operations. When dealing with a large number of histograms, concatenating these histograms and running Sinkhorn's iterations in parallel as matrix-matrix product results in significant speedup that can exploit GPGPU to compare a large number of pairs simultaneously. This makes our approach especially well-suited for large sets of persistence diagrams.

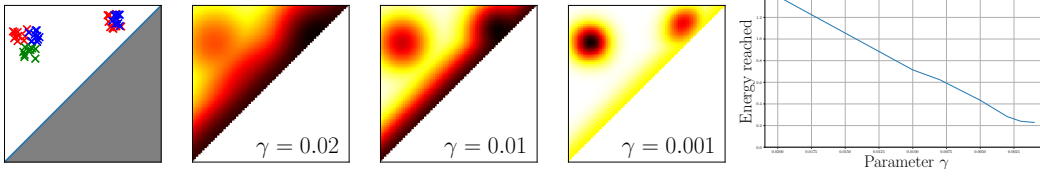

Figure 4: Barycenter estimation for different $\gamma$s with a simple set of 3 PDs (red, blue and green). The smaller the $\gamma$, the better the estimation ($\mathcal{E}$ decreases, note the $\gamma$-axis is flipped on the right plot), at the cost of more iterations in Alg. 2. The mass appearing along the diagonal is a consequence of entropic smoothing: it does not cost much to delete while it increases the entropy of transport plans.

We can now estimate distances between persistence diagrams with Alg. 1 in parallel by performing only $(d \times d)$-sized matrix multiplications, leading to a computational scaling in $d^3$ where $d$ is the grid resolution parameter. Note that a standard stopping threshold in Sinkhorn iteration process is to check the error to marginals $\mathrm{dist}(P, \Pi(\mathbf{a}, \mathbf{b}))$, as motivated by (9).

---

**Algorithm 1** Sinkhorn divergence for persistence diagrams

**Input:** Pairs of PDs $(D_i, D_i')_i$, smoothing parameter $\gamma > 0$, grid step $d \in \mathbb{N}$, stopping criterion, initial $(\mathbf{u}, \mathbf{v})$.
**Output:** Approximation of all $(d_p(D_i, D_i')^p)_i$, upper and lower bounds if wanted.
**init** Cast $D_i, D_i'$ as histograms $\mathbf{a}_i, \mathbf{b}_i$ on a $d \times d$ grid
**while** stopping criterion not reached **do**
    Iterate in parallel (5) $(\mathbf{u}, \mathbf{v}) \mapsto S(\mathbf{u}, \mathbf{v})$ using (11)
**end while**
Compute all $\mathbf{S}_C^\gamma(\mathbf{a}_i + \mathbf{R}\mathbf{b}_i, \mathbf{b}_i + \mathbf{R}\mathbf{a}_i)$ using (12)
**if** Want a upper bound **then**
    Compute $\langle R_i, C \rangle$ in parallel using (14)
**end if**
**if** Want a lower bound **then**
    Compute $\langle (\alpha_t^\gamma)^{c\bar{c}}, \mathbf{a}_i \rangle + \langle (\alpha_t^\gamma)^c, \mathbf{b}_i \rangle$ using (Lucet, 2010)
**end if**

---

## 4 Smoothed barycenters for persistence diagrams

**OT formulation for barycenters.** We show in this section that the benefits of entropic regularization also apply to the computation of barycenters of PDs. As the space of PD is not Hilbertian but only a metric space, the natural definition of barycenters is to formulate them as Fréchet means for the $d_p$ metric, as first introduced (for PDs) in (Mileyko et al., 2011).

**Definition.** *Given a set of persistence diagrams $D_1, \ldots, D_N$, a barycenter of $D_1 \ldots D_N$ is any solution of the following minimization problem:*

$$\underset{\mu \in \mathcal{M}_+(\mathbb{R}_>^2)}{\text{minimize}} \; \mathcal{E}(\mu) := \sum_{i=1}^{N} \mathbf{L}_C(\mu + \mathbf{R}D_i, D_i + \mathbf{R}\mu) \quad (15)$$

*where $C$ is defined as in (8) with $p = 2$ (but our approach adapts easily to any finite $p \geq 1$), and $\mathcal{M}_+(\mathbb{R}_>^2)$ denotes the set of non-negative finite measures supported on $\mathbb{R}_>^2$. $\mathcal{E}(\mu)$ is the* energy *of $\mu$.*

Let $\widehat{\mathcal{E}}$ denotes the restriction of $\mathcal{E}$ to the space of persistence diagrams (finite point measures). Turner et al. (2014) proved the existence of minimizers of $\widehat{\mathcal{E}}$ and proposed an algorithm that converges to a local minimum of the functional, using the Hungarian algorithm as a subroutine. Their algorithm will be referred to as the *B-Munkres Algorithm*. The non-convexity of $\widehat{\mathcal{E}}$ can be a real limitation in practice since $\widehat{\mathcal{E}}$ can have arbitrarily bad local minima (see Lemma 1 in the supplementary material). Note that minimizing $\mathcal{E}$ instead of $\widehat{\mathcal{E}}$ will not give strictly better minimizers (see Proposition 6 in the supplementary material). We then apply entropic smoothing to this problem. This relaxation offers differentiability and circumvents both non-convexity and numerical scalability.

**Entropic smoothing for PD barycenters.** In addition to numerical efficiency, an advantage of smoothed optimal transport is that $a \mapsto \mathbf{L}_C^\gamma(a, b)$ is differentiable. In the Eulerian setting, its gradient is given by centering the vector $\gamma \log(u^\gamma)$ where $u^\gamma$ is a fixed point of the Sinkhorn map (5), see (Cuturi & Doucet, 2014). This result can be adapted to our framework, namely:

**Proposition 5.** *Let $D_1 \ldots D_N$ be PDs, and $(\mathbf{a}_i)_i$ the corresponding histograms on a $d \times d$ grid. The gradient of the functional $\mathcal{E}^\gamma : \mathbf{z} \mapsto \sum_{i=1}^{N} \mathbf{L}_C^\gamma(\mathbf{z} + \mathbf{R}\mathbf{a}_i, \mathbf{a}_i + \mathbf{R}\mathbf{z})$ is given by*

$$\nabla_\mathbf{z} \mathcal{E}^\gamma = \gamma \left( \sum_{i=1}^{N} \log(u_i^\gamma) + \mathbf{R}^T \log(v_i^\gamma) \right) \quad (16)$$

*where $\mathbf{R}^T$ denotes the adjoint operator $\mathbf{R}$ and $(u_i^\gamma, v_i^\gamma)$ is a fixed point of the Sinkhorn map obtained while transporting $\mathbf{z} + \mathbf{R}\mathbf{a}_i$ onto $\mathbf{a}_i + \mathbf{R}\mathbf{z}$.*

As in (Cuturi & Doucet, 2014), this result follows from the envelope theorem, with the added subtlety that $\mathbf{z}$ appears in both terms depending on $u$ and $v$. This formula can be exploited to compute barycenters via gradient descent, yielding Algorithm 2. Following (Cuturi & Doucet, 2014, §4.2), we used a multiplicative update. This is a particular case of mirror descent (Beck & Teboulle, 2003) and is equivalent to a (Bregman) projected gradient descent on the positive orthant, retaining positive coefficients throughout iterations.

As it can be seen in Fig. 4, the barycentric persistence diagrams are smeared.

---

**Algorithm 2** Smoothed approximation of PD barycenter

**Input:** PDs $D_1, \ldots, D_N$, learning rate $\lambda$, smoothing parameter $\gamma > 0$, grid step $d \in \mathbb{N}$.
**Output:** Estimated barycenter $\mathbf{z}$
**Init:** $\mathbf{z}$ uniform measure above the diagonal.
Cast each $D_i$ as an histogram $\mathbf{a}_i$ on a $d \times d$ grid
**while** $\mathbf{z}$ changes **do**
    Iterate $S$ defined in (5) in parallel between all the pairs $(\mathbf{z} + \mathbf{R}\mathbf{a}_i)_i$ and $(\mathbf{a}_i + \mathbf{R}\mathbf{z})_i$, using (11).
    $\nabla := \gamma(\sum_i \log(u_i^\gamma) + \mathbf{R}^T \log(v_i^\gamma))$
    $\mathbf{z} := \mathbf{z} \odot \exp(-\lambda\nabla)$
**end while**
**if** Want energy **then**
    Compute $\frac{1}{N} \sum_i \mathbf{S}_C^\gamma(\mathbf{z} + \mathbf{R}\mathbf{a}_i, \mathbf{a}_i + \mathbf{R}\mathbf{z})$ using (12)
**end if**
Return $\mathbf{z}$

---

If one wishes to recover more spiked diagrams, quantization and/or entropic sharpening (Solomon et al., 2015, §6.1) can be applied, as well as smaller values for $\gamma$ that impact computational speed or numerical stability. We will consider these extensions in future work.

**A comparison with linear representations.** When doing statistical analysis with PDs, a standard approach is to transform a diagram into a finite dimensional vector—in a stable way—and then perform statistical analysis and learning with an Euclidean structure. This approach does not preserve the Wasserstein-like geometry of the diagram space and thus loses the algebraic interpretability of PDs. Fig. 1 gives a qualitative illustration of the difference between Wasserstein barycenters (Fréchet mean) of PDs and Euclidean barycenters (linear means) of persistence images (Adams et al., 2017), a commonly used vectorization for PDs (Makarenko et al., 2016; Zeppelzauer et al., 2016; Obayashi et al., 2018).

## 5 Experiments

All experiments are run with $p = 2$, but would work with any finite $p \geq 1$. This choice is consistent with the work of Turner et al. (2014) for barycenter estimation.

**A large scale approximation.** Iterations of Sinkhorn map (5) yield a transport cost whose value converges to the true transport cost as $\gamma \to 0$ and the number of iterations $t \to \infty$ (Cuturi, 2013). We quantify in Fig. 3 this convergence experimentally using the upper and lower bounds provided in (10) through $t$ and for decreasing $\gamma$. We consider a set of $N = 100$ pairs of diagrams randomly generated with 100 to 150 points in each diagrams, and discretized on a $100 \times 100$ grid. We run Alg. 1 for different $\gamma$ ranging from $10^{-1}$ to $5.10^{-4}$ along with corresponding upper and lower bounds described in (10). For each pair of diagrams, we center our estimates by removing the true distance, so that the target cost is 0 across all pairs. We plot median, top 90% and bottom 10% percentiles for both bounds. Using

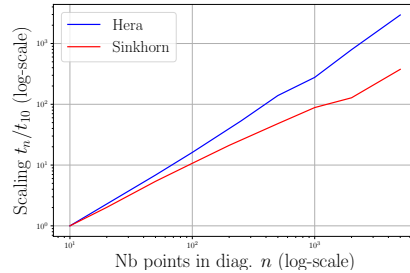

Figure 5: Comparison of scalings of Hera and Sinkhorn (Alg. 1) as the number of points in diagram increases. log-log scale.

the $C$-transform provides a much better lower bound in our experiments. This is however inefficient in practice: despite a theoretical complexity linear in the grid size, the sequential structure of the algorithms described in (Lucet, 2010) makes them unsuited for GPGPU to our knowledge.

We then compare the scalability of Alg. 1 with respect to the number of points in diagrams with that of Kerber et al. (2017) which provides a state-of-the-art algorithm with publicly available code—referred to as *Hera*—to estimate distances between diagrams. For both algorithms, we compute the average time $t_n$ to estimate a distance between two random diagrams having from $n$ to $2n$ points where $n$ ranges from 10 to 5000. In order to compare their scalability, we plot in Fig. 5 the ratio $t_n/t_{10}$ of both algorithms, with $\gamma_n = 10^{-1}/n$ in Alg. 1.

| | | | |
|---|---|---|---|
| (a) Diagram set | (b) B-Munkres | (c) B-Munkres | (d) Alg 2 |

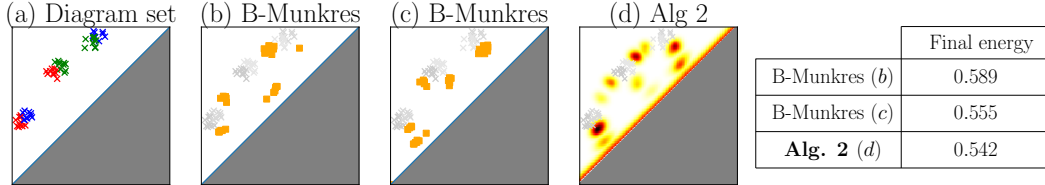

| | Final energy |
|---|---|
| B-Munkres ($b$) | 0.589 |
| B-Munkres ($c$) | 0.555 |
| **Alg. 2** ($d$) | 0.542 |

Figure 7: Qualitative comparison of B-Munkres and our Alg 2. ($a$) Input set of $N = 3$ diagrams with $n = 20$ points each. ($b$) Output of B-Munkres when initialized on the blue diagram (orange squares) and input data (grey scale). ($c$) Output of B-Munkres initialized on the green diagram. ($d$) Output of Alg. 2 on a $100 \times 100$ grid, $\gamma = 5.10^{-4}$, learning-rate $\lambda = 5$, Sinkhorn stopping criterion (distance to marginals): 0.001, gradient descent performed until $|\mathcal{E}(\mathbf{z_{t+1}})/\mathcal{E}(\mathbf{z_t}) - \mathbf{1}| < 0.01$.—As one can see, localization of masses is similar. Initialization of B-Munkres is made on one of the input diagram as indicated in (Turner et al., 2014, Alg. 1), and leads to convergence to different local minima. Our convex approach (Alg. 2) performs better (lower energy). As a baseline, the energy of the naive arithmetic mean of the three diagrams is 0.72.

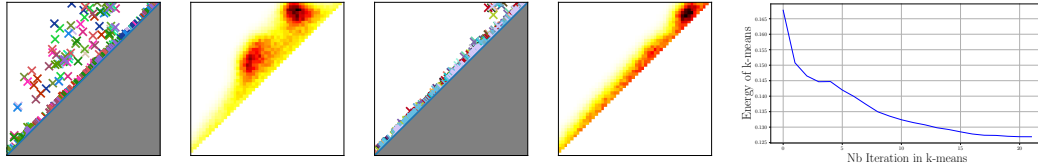

Figure 8: Illustration of our $k$-means algorithm. From left to right: 20 diagrams extracted from *horses* and *camels* plot together (one color for each diagram); the centroid they are matched with provided by our algorithm; 20 diagrams of *head* and *faces*; along with their centroid; decrease of the objective function. Running time depends on many parameters along with the random initialization of $k$-means. As an order of magnitude, it takes from 40 to 80 minutes with this 5000 PD dataset on a P100 GPU.

**Fast barycenters and $k$-means on large PD sets.** We compare our Alg. 2 (referred to as *Sinkhorn*) to the combinatorial algorithm of Turner et al. (2014) (referred to as *B-Munkres*). We use the script `munkres.py` provided on the website of K.Turner for their implementation. We record in Fig. 6 running times of both algorithms on a set of 10 diagrams having from $n$ to $2n$ points, $n$ ranging from 1 to 500, on Intel Xeon 2.3 GHz (CPU) and P100 (GPU, Sinkhorn only). When running Alg. 2, the gradient descent is performed until $|\mathcal{E}(\mathbf{z}_{t+1})/\mathcal{E}(\mathbf{z}_t) - 1| < 0.01$, with $\gamma = 10^{-1}/n$ and $d = 50$. Our experiment shows that Alg. 2 drastically outperforms *B-Munkres* as the number of points $n$ increases. We interrupt *B-Munkres* at $n = 30$, after which computational time becomes an issue.

Aside the computational efficiency, we highlight the benefits of operating with a convex formulation in Fig. 7. Due to non-convexity, the B-Munkres algorithm is only guaranteed to converge to a local minima, and its output depends on initialization. We illustrate on a toy set of $N = 3$ diagrams how our algorithm avoids local minima thanks to the Eulerian approach we take.

We now merge Alg. 1 and Alg. 2 in order to perform unsupervised clustering via $k$-means on PDs. We work with the 3D-shape database provided by Sumner & Popović and generate diagrams in the same way as in (Carrière et al., 2015),

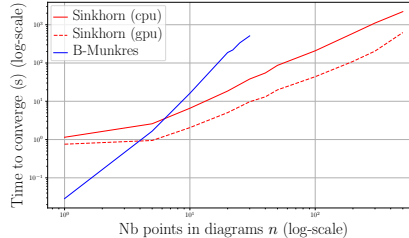

Figure 6: Average running times for B-Munkres *(blue)* and Sinkhorn *(red)* algorithms (log-log scale) to average 10 PDs.

working in practice with 5000 diagrams with 50 to 100 points each. The database contains 6 classes: `camel`, `cat`, `elephant`, `horse`, `head` and `face`. In practice, this unsupervised clustering algorithm detects two main clusters: faces and heads on one hand, camels and horses on the other hand are systematically grouped together. Fig. 8 illustrates the convergence of our algorithm and the computed centroids for the aforementioned clusters.

## 6 Conclusion

In this work, we took advantage of a link between PD metrics and optimal transport to leverage and adapt entropic regularization for persistence diagrams. Our approach relies on matrix manipulations rather than combinatorial computations, providing parallelization and efficient use of GPUs. We provide bounds to control approximation errors. We use these differentiable approximations to estimate barycenters of PDs significantly faster than existing algorithm, and showcase their application by clustering thousand diagrams built from real data. We believe this first step will open the way for new statistical tools for TDA and ambitious data analysis applications of persistence diagrams.

**Acknowledgments.** We thank the anonymous reviewers for the fruitful discussion. TL was supported by the AMX, École polytechnique. MC acknowledges the support of a Chaire d'Excellence de l'Idex Paris-Saclay.

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
