[Supplementary Material]

# 7 Supplementary material

## 7.1 Omitted proofs from Section 3

**Diagram metrics as optimal transport:** We recall that we consider $D_1 = \sum_{i=1}^{n_1} \delta_{x_i}$ and $D_2 = \sum_{j=1}^{n_2} \delta_{y_j}$ two persistence diagrams with respectively $n_1$ points $x_1 \ldots x_{n_1}$ and $n_2$ points $y_1 \ldots y_{n_2}$, $p \geq 1$, and $C$ is the cost matrix with block structure

$$C = \begin{pmatrix} \widehat{C} & u \\ v^T & 0 \end{pmatrix} \in \mathbb{R}^{(n_1+1)\times(n_2+1)},$$

*Proof of Prop. 1.* Let $n = n_1 + n_2$ and $\mu = D_1 + \mathbf{R}D_2, \nu = D_2 + \mathbf{R}D_1$. Since $\mu, \nu$ are point measures, that is discrete measures of same mass $n$ with integer weights at each point of their support, finding $\inf_{P \in \Pi(\mu,\nu)} \langle P, C \rangle$ is an assignment problem of size $n$ as introduced in §2. It is equivalent to finding an optimal matching $P \in \Sigma_n$ representing some permutation $\sigma \in \mathfrak{S}_n$ for the cost matrix $\widetilde{C} \in \mathbb{R}^{n \times n}$ built from $C$ by repeating the last line $u$ in total $n_1$ times, the last column $v$ in total $n_2$ times, and replacing the lower right corner $0$ by a $n_1 \times n_2$ matrix of zeros. The optimal $\sigma$ defines a partial matching $\zeta$ between $D_1$ and $D_2$, defined by $(x_i, y_j) \in \zeta$ iff $j = \sigma(i)$, $1 \leq i \leq n_1, 1 \leq j \leq n_2$. Such pairs of points induce a cost $\|x_i - y_j\|^p$, while other points $s \in D_1 \cup D_2$ (referred to as unmatched) induce a cost $\|s - \pi_\Delta(s)\|^p$. Then:

$$\begin{aligned}
\mathbf{L}_C(\mu, \nu) &= \min_{P \in \Sigma_n} \langle \widetilde{C}, P \rangle \\
&= \min_{\sigma \in \mathfrak{S}_n} \sum_{i=1}^{n} \widetilde{C}_{i\sigma(i)} \\
&= \min_{\zeta \in \Gamma(D_1, D_2)} \sum_{(x_i, y_j) \in \zeta} \|x_i - y_j\|^p + \sum_{\substack{s \in D_1 \cup D_2 \\ s \text{ unmatched by } \zeta}} \|s - \pi_\Delta(s)\|^p \\
&= d_p(D_1, D_2)^p.
\end{aligned}$$

$\square$

**Error control due to discretization:** Let $D_1, D_2$ be two diagrams and $\mathbf{a}, \mathbf{b}$ their respective representations as $d \times d$ histograms. For two histograms, $\mathbf{L}_C(\mathbf{a} + \mathbf{R}\mathbf{b}, \mathbf{b} + \mathbf{R}\mathbf{a}) = d_p(D_1' + \mathbf{R}D_2', D_2' + \mathbf{R}D_1')$ where $D_1', D_2'$ are diagrams deduced from $D_1, D_2$ respectively by moving any mass located at $(x, y) \in \mathbb{R}_>^2 \cap [0,1]^2$ to $\left(\frac{\lfloor xd \rfloor}{d}, \frac{\lfloor yd \rfloor}{d}\right)$, inducing at most an error of $\frac{1}{d}$ for each point. We identify $\mathbf{a}, \mathbf{b}$ and $D_1', D_2'$ in the following. We recall that $d_p(\cdot, \cdot)$ is a distance over persistence diagrams and thus satisfy triangle inequality, leading to:

$$|d_p(D_1, D_2) - \mathbf{L}_C(\mathbf{a} + \mathbf{R}\mathbf{b}, \mathbf{b} + \mathbf{R}\mathbf{a})^{\frac{1}{p}}| \leq d_p(D_1, D_1') + d_p(D_2, D_2')$$

Thus, the error made is upper bounded by $\frac{1}{d}(|D_1|^{\frac{1}{p}} + |D_2|^{\frac{1}{p}})$.

**Propositions 2, 3, 4:** We keep the same notations as in the core of the article and give details regarding the iteration schemes provided in the paper.

*Proof of prop 2.* Given an histogram $\mathbf{u} \in \mathbb{R}^{d \times d}$ and a mass $u_\Delta \in \mathbb{R}_+$, one can observe that (see below):

$$\widehat{K}\mathbf{u} = \mathbf{k}(\mathbf{k}\mathbf{u}^T)^T. \tag{17}$$

In particular, the operation $\mathbf{u} \mapsto \widehat{K}\mathbf{u}$ can be perform by only manipulating matrices in $\mathbb{R}^{d \times d}$. Indeed, observe that:

$$\widehat{K}_{ij,kl} = e^{-(i-k)^2/\gamma} e^{-(j-l)^2/\gamma} = \mathbf{k}_{ik}\mathbf{k}_{jl},$$

so we have:

$$(\widehat{K}\mathbf{u})_{i,j} = \sum_{k,l} K_{ij,kl}\mathbf{u}_{k,l}$$

$$= \sum_{k,l} \mathbf{k}_{ik}\mathbf{k}_{jl}\mathbf{u}_{k,l} = \sum_k \mathbf{k}_{ik}\sum_l \mathbf{k}_{jl}\mathbf{u}_{kl}$$

$$= \sum_k \mathbf{k}_{ik}(\mathbf{k}\mathbf{u}^T)_{jk} = (\mathbf{k}(\mathbf{k}\mathbf{u}^T)^T)_{i,j}.$$

Thus we have in our case:

$$K(\mathbf{u}, u_\Delta) = (\widehat{K}\mathbf{u} + u_\Delta\mathbf{k}_\Delta, \langle\mathbf{u}, \mathbf{k}_\Delta\rangle + u_\Delta)$$

where $\langle a, b\rangle$ designs the Froebenius dot product between two histograms $a, b \in \mathbb{R}^{d\times d}$. Note that these computations only involves manipulation of matrices with size $d \times d$. $\qquad\square$

*Proof of prop 3.*

$$\langle\mathrm{diag}(\overrightarrow{\mathbf{u}})\widehat{K}\mathrm{diag}(\overrightarrow{\mathbf{v}}), \widehat{C}\rangle = \sum_{ijkl} \mathbf{u}_{ij}\mathbf{k}_{ik}\mathbf{k}_{jl}[\mathbf{c}_{ik} + \mathbf{c}_{jl}]\mathbf{v}_{kl}$$

$$= \sum_{ijkl} \mathbf{u}_{ij}\left([\mathbf{k}_{ik}\mathbf{c}_{ik}]\mathbf{k}_{jl}\mathbf{v}_{kl} + \mathbf{k}_{ik}[\mathbf{k}_{jl}\mathbf{c}_{jl}]\mathbf{v}_{kl}\right)$$

$$= \sum_{ij} \mathbf{u}_{ij}\sum_{kl}\left(\mathbf{m}_{ik}\mathbf{k}_{jl}\mathbf{v}_{kl} + \mathbf{k}_{ik}\mathbf{m}_{jl}\mathbf{v}_{kl}\right)$$

Thus, we finally have:

$$\langle\mathrm{diag}(\overrightarrow{\mathbf{u}})\widehat{K}\mathrm{diag}(\overrightarrow{\mathbf{v}}), \widehat{C}\rangle = \|\mathbf{u}\odot\left(\mathbf{m}(\mathbf{k}\mathbf{v}^T)^T + \mathbf{k}\mathbf{m}\mathbf{v}^T]^T\right)\|_1$$

And finally, taking the $\{\Delta\}$ bin into considerations,

$$\langle\mathrm{diag}(\overrightarrow{\mathbf{u}}, u_\Delta)K\mathrm{diag}(\overrightarrow{\mathbf{v}}, v_\Delta), C\rangle = \langle\begin{pmatrix}\mathrm{diag}(\overrightarrow{\mathbf{u}})\widehat{K}\mathrm{diag}(\overrightarrow{\mathbf{v}}) & v_\Delta(\overrightarrow{\mathbf{u}}\odot\overrightarrow{\mathbf{k}}_\Delta) \\ u_\Delta(\overrightarrow{\mathbf{v}}^T\odot\overrightarrow{\mathbf{k}}_\Delta^T) & u_\Delta v_\Delta\end{pmatrix}, \begin{pmatrix}\widehat{C} & \overrightarrow{\mathbf{c}}_\Delta \\ \overrightarrow{\mathbf{c}}_\Delta^T & 0\end{pmatrix}\rangle$$

$$= \langle\mathrm{diag}(\overrightarrow{\mathbf{u}})\widehat{K}\mathrm{diag}(\overrightarrow{\mathbf{v}}), \widehat{C}\rangle + u_\Delta\langle\mathbf{v}, \mathbf{k}_\Delta\odot\mathbf{c}_\Delta\rangle + v_\Delta\langle\mathbf{u}, \mathbf{k}_\Delta\odot\mathbf{c}_\Delta\rangle$$

Remark: First term correspond to the cost of effective mapping (point to point) and the two others to the mass mapped to the diagonal. $\qquad\square$

To address the last proof, we recall below the `rounding_to_feasible` algorithm introduced by Altschuler et al.; $r(P)$ and $c(P)$ denotes respectively the first and second marginal of a matrix $P$.

---

**Algorithm 3** Rounding algorithm of Altschuler et al. (2017)

---

1: **Input:** $P \in \mathbb{R}^{d\times d}$, desired marginals $r, c$.
2: **Output:** $F(P) \in \Pi(r,c)$ close to $P$.
3: $X = \min\left(\frac{r}{r(P)}, 1\right) \in \mathbb{R}^d$
4: $P' = \mathrm{diag}(X)P$
5: $Y = \min\left(\frac{c}{c(P')}, 1\right) \in \mathbb{R}^d$
6: $P'' = P'\mathrm{diag}(Y)$
7: $e_r = r - r(P''), e_c = c - c(P'')$
8: **return** $F(P) := P'' + e_r e_c^T/\|e_c\|_1$

---

*Proof of prop 4.* By straightforward computations, the first and second marginals of $P_t^\gamma = \mathrm{diag}(\overrightarrow{\mathbf{u}})K\mathrm{diag}(\overrightarrow{\mathbf{v}})$ are given by:

$$\left(\sum_{kl}\mathbf{u}_{ij}K_{ij,kl}\mathbf{v}_{kl}\right)_{ij} = \mathbf{u}\odot(K\mathbf{v}), \qquad \left(\sum_{ij}\mathbf{u}_{ij}K_{ij,kl}\mathbf{v}_{kl}\right)_{kl} = (\mathbf{u}K)\odot\mathbf{v}.$$

Observe that $K\mathbf{v}$ and $\mathbf{u}K$ can be computed using Proposition 2.

Now, the transport cost computation is:

$$\langle F(P_t^\gamma), C\rangle = \langle \mathrm{diag}(X)P_t^\gamma \mathrm{diag}(Y), C\rangle + \langle e_r e_c^T / \|e_c\|_1, C\rangle$$

$$= \langle \mathrm{diag}(X \odot \mathbf{u})K\mathrm{diag}(Y \odot \mathbf{v}), C\rangle + \frac{1}{\|e_c\|_1} \sum_{ijkl} (e_r)_{ij}(e_c)_{kl}[\mathbf{c}_{ik} + \mathbf{c}_{jl}]$$

The first term is the transport cost induced by a rescaling of $\mathbf{u}, \mathbf{v}$ and can be computed with Prop 3. Consider now the second term. Without considering the additional bin $\{\Delta\}$, we have:

$$\sum_{ijkl}(e_r)_{ij}(e_c)_{kl}[\mathbf{c}_{ik} + \mathbf{c}_{jl}] = \sum_{ijl}(e_r)_{ij}\sum_k \mathbf{c}_{ik}(e_c)_{kl} + \sum_{ijk}(e_r)_{ij}\sum_l \mathbf{c}_{jl}(e_c)_{kl}$$

$$= \sum_{ijl}(e_r)_{ij}(\mathbf{c}e_c)_{il} + \sum_{ijk}(e_r)_{ij}(\mathbf{c}e_c^T)_{jk}$$

$$= \|e_r^T \mathbf{c}e_c\|_1 + \|e_r \mathbf{c}e_c^T\|_1,$$

so when we consider our framework (with $\{\Delta\}$), it comes:

$$\langle \begin{pmatrix} e_r \\ (e_r)_\Delta \end{pmatrix} \cdot \begin{pmatrix} e_c & (e_c)_\Delta \end{pmatrix}, C\rangle = \langle \begin{pmatrix} e_r e_c^T & (e_c)_\Delta e_r \\ (e_r)_\Delta e_c^T & (e_r)_\Delta (e_c)_\Delta \end{pmatrix}, \begin{pmatrix} \widehat{C} & \vec{\mathbf{c}}_\Delta \\ \vec{\mathbf{c}}_\Delta^T & 0 \end{pmatrix}\rangle$$

$$= \langle e_r e_c^T, \widehat{C}\rangle + (e_c)_\Delta \langle e_r, \mathbf{c}_\Delta\rangle + (e_r)_\Delta \langle e_c, \mathbf{c}_\Delta\rangle .$$

Putting things together finally proves the claim. $\qquad\square$

## 7.2 Omitted proofs from Section 4

We first observe that $\mathcal{E}$ does not have local minimum (while $\widehat{\mathcal{E}}$ does). For $x \in \mathbb{R}^2_> \cup \{\Delta\}$, we extend the Euclidean norm by $\|x - \Delta\|$ the distance from $x$ to its orthogonal projection onto the diagonal $\pi_\Delta(x)$. In particular, $\|\Delta - \Delta\| = 0$. We denote by $c$ the corresponding cost function (continuous analogue of the matrix $C$ defined in (8)).[1]

**Proposition** (Convexity of $\mathcal{E}$). *For any two measures $\mu, \mu' \in \mathcal{M}_+(\mathbb{R}^2_>)$ and $t \in (0, 1)$, we have:*

$$\mathcal{E}((1-t)\mu + t\mu') \leq (1-t)\mathcal{E}(\mu) + t\mathcal{E}(\mu') \tag{18}$$

*Proof.* We denote by $\alpha_i, \beta_i$ the dual variables involved when computing the optimal transport plan between $(1-t)\mu + t\mu' + \mathbf{R}D_i$ and $D_i + \mathbf{R}((1-t)\mu + t\mu')$. Note that maximum are taken over the set $\alpha_i, \beta_i | \alpha_i \oplus \beta_i \leq c$ (with $\alpha \oplus \beta : (x, y) \mapsto \alpha(x) + \beta(y)$):

$$\mathcal{E}((1-t)\mu + t\mu') = \frac{1}{n}\sum_{i=1}^n \mathbf{L}_c((1-t)\mu + t\mu' + \mathbf{R}D_i, D_i + (1-t)\mathbf{R}\mu + t\mathbf{R}\mu')$$

$$= \frac{1}{n}\sum_{i=1}^n \max\{\langle \alpha_i, (1-t)\mu + t\mu' + \mathbf{R}D_i\rangle + \langle \beta_i, D_i + (1-t)\mathbf{R}\mu + t\mathbf{R}\mu'\rangle\}$$

$$= \frac{1}{n}\sum_{i=1}^n \max\{(1-t)\left(\langle \alpha_i, \mu + \mathbf{R}D_i\rangle + \langle \beta_i, D_i + \mathbf{R}\mu\rangle\right) +$$

$$t\left(\langle \alpha_i, \mu' + \mathbf{R}D_i\rangle + \langle \beta_i, D_i + \mathbf{R}\mu'\rangle\right)\}$$

$$\leq \frac{1}{n}\sum_{i=1}^n (1-t)\max\{\langle \alpha_i, \mu + \mathbf{R}D_i\rangle + \langle \beta_i, D_i + \mathbf{R}\mu\rangle\}$$

$$+ t\max\{\langle \alpha_i, \mu' + \mathbf{R}D_i\rangle + \langle \beta_i, D_i + \mathbf{R}\mu'\rangle\}$$

$$= (1-t)\frac{1}{n}\sum_{i=1}^n \mathbf{L}_c(\mu + \mathbf{R}D_i, D_i + \mathbf{R}\mu) + t\frac{1}{n}\sum_{i=1}^n \mathbf{L}_c(\mu' + \mathbf{R}D_i, D_i + \mathbf{R}\mu')$$

$$= (1-t)\mathcal{E}(\mu) + t\mathcal{E}(\mu').$$

$\square$

**Tightness of the relaxation.** The following result states that the minimization problem (15) is a tight relaxation of the problem considered by Turner et al. in sense that global minimizers of $\widehat{\mathcal{E}}$ (which are, by definition, persistence diagrams) are (global) minimizers of $\mathcal{E}$.

**Proposition 6.** *Let $D_1, \ldots, D_N$ be a set of persistence diagrams. Diagram $D_i$ has mass $m_i \in \mathbb{N}$, while $m_{\mathrm{tot}} = \sum m_i$ denotes the total mass of the dataset. Consider the* normalized dataset *$\widehat{D_1}, \ldots, \widehat{D_N}$ defined by $\widehat{D_i} := D_i + (m_{\mathrm{tot}} - m_i)\delta_\Delta$. Then the functional*

$$\mathcal{G} : \mu \mapsto \frac{1}{N} \sum_{i=1}^N \mathbf{L}_c(\mu + (m_{\mathrm{tot}} - |\mu|)\delta_\Delta, \widehat{D_i}) \tag{19}$$

*where $\mu \in \{\mathcal{M}_+(\mathbb{R}^2_>) : \max_i m_i \le |\mu| \le m_{\mathrm{tot}}\}$ has the same minimizers as (15).*

**Corollary** (Properties of barycenters for PDs)**.** *Let $\mu^*$ be a minimizer of (15). Then $\mu^*$ satisfies:*

 *(i) (Carlier et al., 2015) Localization: $x \in \mathrm{supp}(\mu^*) \Rightarrow x$ minimizes $z \mapsto \sum_{i=1}^n \|x_i - z\|_2^2$ for some $x_i \in \mathrm{supp}(\widehat{D_i})$. This function admit a unique minimizer in $\mathbb{R}^2_> \cup \{\Delta\}$, thus the support of $\mu^*$ is discrete.*

 *(ii) $\mathcal{G}$ admits persistence diagrams (that is finite discrete measures with integer masses) as minimizers (so does $\mathcal{E}$).*

We introduce an intermediate function $\mathcal{F}$, which appears to have same minimizers as $\mathcal{E}$ and $\mathcal{G}$, which will allow us to conclude that $\mathcal{E}$ and $\mathcal{G}$ have the same set of minimizers.

**Proposition.** *Let $\mu^* \in \mathcal{M}_+(\mathbb{R}^2_>)$ be a minimizer of $\mathcal{E}$ and $(P_i)_i$ the corresponding optimal transport plans. Then for all $i$, $P_i$ fully transports $D_i$ onto $\mu^*$ (i.e. $P_i(x, \Delta) = 0$ for any $x \in \mathrm{supp}(D_i)$). In particular, $|\mu^*| \ge \max m_i$ and $\mathcal{E}$ has the same minimizers as:*

$$\mathcal{F}(\mu) := \frac{1}{N} \sum_{i=1}^N \mathbf{L}_c(\mu, D_i + (|\mu| - m_i)\delta_\Delta) \tag{20}$$

*where $\mu \in \mathcal{M}_+(\mathbb{R}^2_>)$ and satisfies $|\mu| \ge \max m_i$*

*Proof.* Fix $i \in \{1 \ldots N\}$. Let $P_i$ be an optimal transport plan between $\mu^* + m_i\delta_\Delta$ and $D_i + |\mu^*|\delta_\Delta$. Let $x \in \mathrm{supp}(D_i)$. Assume that there is a fraction of mass $t > 0$ located at $x$ that is transported to the diagonal $\Delta$.

Consider the measure $\mu' := \mu^* + t\delta_{x'}$, where $x' = \frac{x + (N-1)\pi_\Delta(x)}{N}$. We now define the transport plan $P_i'$ which is adapted from $P_i$ by transporting the previous mass to $x'$ instead of $\Delta$ (inducing a cost $t\|x - x'\|^2$ instead of $t\|x - \Delta\|^2$). Extend all other optimal transport plans $(P_j)_{j \ne i}$ to $P_j'$ by transporting the mass $t$ located at $x'$ in $\mu'$ to the diagonal $\Delta$ (inducing a total cost $(N-1)t\|x' - \Delta\|^2$), and everything else remains unchanged. One can observe that the new $(P_j')_j$ are admissible transport plans from $\mu' + m_j\delta_\Delta$ to $D_j + |\mu'|\delta_\Delta$ (respectively) inducing an energy $\mathcal{E}(\mu')$ strictly smaller than $\mathcal{E}(\mu^*)$, leading to a contradiction since $\mathcal{E}(\mu^*)$ is supposed to be optimal.

To prove equivalence between the two problems considered (in the sense that they have the same minimizers), we introduce $\mu_\mathcal{E}^*$ and $\mu_\mathcal{F}^*$ which are minimizers of $\mathcal{E}$ and $\mathcal{F}$ respectively. Note that the existence of minimizers is given by standard arguments in optimal transport theory (lower semi-continuity of $\mathcal{E}, \mathcal{F}, \mathcal{G}$ and relative compactness of minimizing sequences, see for example (Agueh & Carlier, 2011, Prop. 2.3)). We first observe that $\mathcal{E}(\mu) \le \mathcal{F}(\mu)$ for all $\mu$ (adding the same amount of mass on the diagonal can only decrease the optimal transport cost).

This allows us to write:

$$\begin{aligned}
\mathcal{F}(\mu_\mathcal{E}^*) = \mathcal{E}(\mu_\mathcal{E}^*) && \text{We can remove } m_i\delta_\Delta \text{ from both sides} \\
\le \mathcal{E}(\mu_\mathcal{F}^*) && \text{since } \mu_\mathcal{E}^* \text{ is a minimizer of } \mathcal{E} \\
\le \mathcal{F}(\mu_\mathcal{F}^*) && \text{since } \mathcal{E}(\mu) \le \mathcal{F}(\mu) \\
\le \mathcal{F}(\mu_\mathcal{E}^*) && \text{since } \mu_\mathcal{F}^* \text{ is a minimizer of } \mathcal{F}
\end{aligned}$$

Hence, all these inequalities are actually equalities, thus minimizers of $\mathcal{E}$ are minimizers of $\mathcal{F}$ and vice-versa. $\square$

We can now prove that $\mathcal{F}$ as the same minimizers as $\mathcal{G}$ which will finally prove Proposition 6.

*Proof of Proposition 6.* Let $\mu_{\mathcal{G}}^*$ be a minimizer of $\mathcal{G}$. Consider $\mu_\Delta := (m_{\text{tot}} - |\mu_{\mathcal{G}}^*|)\delta_\Delta$. We observe that $\mu_\Delta$ is always transported on $\{\Delta\}$ (inducing a cost of 0) for each of the transport plan $P_i \in \Pi(\mu_{\mathcal{G}}^* + \mu_\Delta, \widehat{D_i})$ for minimality considerations (as in previous proof). Observe also (as in previous proof) that $\mathcal{G}(\mu) \leq \mathcal{F}(\mu)$ for any measure $\mu$, yielding:

$$
\begin{aligned}
\mathcal{G}(\mu_{\mathcal{G}}^*) = \mathcal{F}(\mu_{\mathcal{G}}^*) && \text{remove } \mu_\Delta \text{ from both sides} \\
\geq \mathcal{F}(\mu_{\mathcal{F}}^*) && \text{since } \mu_{\mathcal{F}}^* \text{ is a minimizer of } \mathcal{F} \\
\geq \mathcal{G}(\mu_{\mathcal{F}}^*) && \text{since } \mathcal{G}(\mu) \leq \mathcal{F}(\mu) \\
\geq \mathcal{G}(\mu_{\mathcal{G}}^*) && \text{since } \mu_{\mathcal{G}}^* \text{ is a minimizer of } \mathcal{G}
\end{aligned}
$$

This implies that minimizers of $\mathcal{G}$ are minimizers of $\mathcal{F}$ (and thus of $\mathcal{E}$) and conversely. $\qquad\square$

**Details for Corollary of Proposition 6**

(i) Given $N$ diagrams $D_1 \ldots D_N$ and $(x_1 \ldots x_N) \in \text{supp}(\widehat{D_1}) \times \cdots \times \text{supp}(\widehat{D_N})$, among which $k$ of them are equals to $\Delta$, on can easily observe (this is mentioned in Turner et al. (2014)) that $z \mapsto \sum_{i=1}^{N} \|z - x_i\|_2^2$ admits a unique minimizer $x^* = \frac{(N-k)\overline{x} + k\pi_\Delta(\overline{x})}{N}$, where $\overline{x}$ is the arithmetic mean of the $(N-k)$ non-diagonal points in $x_1 \ldots x_N$.

The localization property (see §2.2 of Carlier et al. (2015)) states that the support of any barycenter is included in the set $S$ of such $x^*$s which is finite, proving in particular that barycenters of $\widehat{D_1} \ldots \widehat{D_N}$ have a discrete support included in some known set. Note that a similar result is also mentioned in (Anderes et al., 2016).

(ii) As a consequence of previous point, one can describe a barycenter of $\widehat{D_1} \ldots \widehat{D_N}$ as a vector of weight $w \in \mathbb{R}_+^s$, where $s$ is the cardinality of $S$ and cast the barycenter problem as a Linear Programming (LP) one (see for example §3.2 in (Anderes et al., 2016) or §2.3 and 2.4 in (Carlier et al., 2015)). More precisely, the problem is equivalent to:

$$
\underset{w \in \mathbb{R}_+^s}{\text{minimize}}\, w^T c
$$

$$
\text{s.t.} \forall i = 1 \ldots N, A_i w = b_i
$$

Here, $c \in \mathbb{R}^s$ is defined as $c_j = \sum_{k=1}^{N} \|x_j^* - x_{k,j}\|_2^2$, where $x_j^*$ is the mean (as defined above) associated to $(x_{k,j})_{k=1}^{N}$. The constraints correspond to marginals constraints: $b_i$ is the weight vector associated to $\widehat{D_i}$ on each point of its support. Integer Linear Programming (see (Schrijver, 1998)) allows to conclude that among optimal $w$, some of them have integer coordinates.

**Bad local minima of $\widehat{\mathcal{E}}$.** The following lemma illustrate specific situation which lead algorithms proposed by Turner et al. to get stuck in bad local minima.

**Lemma 1.** *For any $\kappa \geq 1$, there exists a set of diagrams such that $\widehat{\mathcal{E}}$ admits a local minimizer $D_{\text{loc}}$ satisfying:*

$$
\widehat{\mathcal{E}}(D_{\text{loc}}) \geq \kappa \widehat{\mathcal{E}}(D_{\text{opt}})
$$

*where $D_{\text{opt}}$ is a global minimizer. Furthermore, there exist sets of diagrams so that the B-Munkres algorithm always converges to such a local minimum when initialized with one of the input diagram.*

*Proof.* We consider the configuration of Fig. 9a where we consider two diagrams with 1 point (blue and green diagram) and their correct barycenter (red diagram) along with the orange diagram (2 points). It is easy to observe that when restricted to the space of persistence diagram, the orange diagram is a minimizer of the function $\widehat{\mathcal{E}}$ (in which the algorithm could get stuck if initialized poorly). It achieves an energy of $\frac{1}{2}((\frac{1}{2} + \frac{1}{2})^2 + (\frac{1}{2} + \frac{1}{2})^2) = 1$ while the red diagram achieves an energy of $\frac{1}{2}(\sqrt{\epsilon}^2 + \sqrt{\epsilon}^2) = \epsilon$. This example proves that there exist configurations of diagrams so that $\widehat{\mathcal{E}}$ has arbitrary bad local minima.

One could argue that when initialized to one of the input diagram (as suggested in (Turner et al., 2014)), the algorithm will not get stuck to the orange diagram. Fig. 9b provide a configuration

involving three diagrams with two points each where the algorithm will always get stuck in a bad local minimum when initialized with any of the three diagrams. The analysis is similar to previous statement.

(a) Example of arbitrary bad local minima of $\widehat{\mathcal{E}}$. Blue point and green point represent our two diagrams of interest. Red point is a global minimizer of $\widehat{\mathcal{E}}$. The two orange points give a diagram which is a local minimizer of $\widehat{\mathcal{E}}$ achieving an energy arbitrary higher (relatively) than the one of the red diagram (as $\epsilon$ goes to 0).

(b) Failing configuration for B-Munkres algorithm. Three diagrams (red, blue, green) along with the output of Turner et al algorithm (purple) when initialized on the green diagram (we have a similar result by symmetry when initialized on any other diagram).

Figure 9: Example of simple configurations in which the B-Munkres algorithm will converge to arbitrarily bad local minima

$\square$

## Footnotes

[1] Optimal transport between non-discrete measures was not introduced in the core of this article for the sake of concision. It is a natural extension of notions introduced in §2 (distances, primal and dual problems, barycenters). We refer the reader to (Santambrogio, 2015; Villani, 2008) for more details.