[Reviews · NeurIPS 2018]

Reviewer 1



This paper proposes a new method for the clustering of persistence diagrams using recent techniques in optimal transport. The problem is quite important; clustering provides a sensible way to group data according to their topological characterizations. It is also very challenging due to the Wasserstein distance between the persistence diagrams. This paper proposes to (1) approximate the Wasserstein distance between diagrams using the regularized optimal transport, and (2) treat the computation of the Frechet means as another optimal transport problem, and find the optimal solution using gradient descent. Several major technical challenges are addressed, include: 1) the Wasserstein distance may involve matching points with the a diagonal line. 2) estimating tighter bounds for the convergence stopping criterion. 3) a convolution based algorithm to improve the computation efficiency. The proposed method is compared with the state-of-the-art (Hera) and is shown to be more efficient. The problem addressed here is to compute the average of a set of diagrams. It is different from computing pairwise distance or kernel. A good tool for this task will lead to efficient clustering of persistence diagrams. This can be the crucial first step for analysis when direct supervised training on persistence diagrams is not effective (which is often the case in practice). The main advantage of the proposed method is it can be implemented in a series of matrix multiplications, which can be efficiently executed using tensorflow + GPU (for the first time, I believe). Other methods, such as B-Munkres and Hera, solve the problem as a discrete optimization task; there is no (obvious) way to implement them in GPU/multi-core. (Hope this answers a question of R4.) I personally think the GPU implementation is extremely important. To really incorporate topological information in deep learning, we need to run topological computations repeatedly during the training (after every epoch, or even every batch). For this level of computational demand, an algorithm that finishes in seconds is already too slow. The paper is well written. A few additional suggestions/comments for improvements: * The technical part is a bit dense. To make it more accessible to readers, it might be necessary to provide some overview of the method in the beginning, especially regarding how the two parts (sec 3 and 4) are connected. * It is not clear to me whether the method allows multiple smoothed barycenters, as Turner’s algorithm does. If so, how? * In Fig 4, it might be necessary to also report the error. For distance computation, Hera is supposed to produce the true distance. But Sinkhorn will be stoped when the error is tolerable, right? * In Fig 5, it will be useful to report the variance achieved by B-Munkres and Sinkhorn respectively, for reference. It is also necessary to show the smeared and the spiked (post processed) barycenter diagrams, as well as the output of B-Munkres, for quantitative evaluations. * Typos: L138: “even but one”. L272: GPGPU. -- Final suggestions: This paper has solid contribution and worth publishing in NIPS. But to be better appreciated by the general learning community, and even many people familiar with TDA, there are certain additional improvements (in terms of presentation) that could be added to the final version. 1) Different computational problems for TDA have been addressed in the past. It would be helpful to provide more contextual information of the addressed problem (clustering of PDs), e.g., how it is related but different from the computation of persistence diagrams, bottleneck/Wasserstein distance, kernels & simple vectorization techniques for supervised learning. A diagram could indeed be very illustrative. During discussion, we (reviewers and AC) feel that to better demonstrate the necessity of the proposed method. It will be very useful to also refer to and (qualitatively) compare with simple vectorization (persistence images)+average or the average of landscapes. These methods definitely have computational advantages, but they will fail to produce meaningful ("sharp") average diagrams (for better interpretations), compared with Wasserstein-distance-based methods.

Reviewer 2



UPDATE: Thanks to the authors for their responses. I appreciate the time they took to provide clarifications and run an additional experiment. ORIGINAL REVIEW: The authors use the theory of optimal transport, particularly Eulerian mass transport with entropic approximation and inclusion of a “sink” bin to perform large scale computation of means and clusterings of persistence diagrams. Along the way, they also provide other contributions such as sinkhorn divergence for PDs as well as computation of PD barycenters. The key contribution seems to be the way that the authors have handled the diagonal points during the computations. The paper is well-written and fairly easy to follow with sufficient background in topology and OT, and I enjoyed reading the paper. Here are my comments: 1. The point that is not clear to me is claiming equal number of points in the two PDs after including bins for the diagonals, but the mathematical expressions showing a different thing. For example, in (8), C is of size (n_1+1) x (n_2+1), but line 120 speaks about equal number of points. 2. In lines 204-205, the histograms are R^(dxd+1) --- the number of diagonal bins is just 1 with a mass equal to that in the other PD. In this case, how can there the transport problem be the same as matching? Since the bin for the diagonal has a much larger mass than other bins, the transport to and from diagonals will not be just one to one matching, right? Please clarify this. 3. Why is the gradient step in Alg. 2 in the exp space? Why is it not simply z = z-\lambda \Delta? Is it because of the property given in lines 239-241? 4. The experiments have covered 2 cases: (a) computing distances between large PDs, (b) computing barycenters and k means for lots of small PDs (more realistic case). It will be good to see a few more datasets. Also, can you demonstrate the quality of your barycenters with respect to B-Munkres, empirically? May be you can compute the distance between the centroid PDs computed by B-Munkres and your centroid approximation? 5. Are the experiments only done for p=2 in d_p(D_1, D_2)? What about p=1? In experiments, p=1 are sometimes preferred because of their enhanced robustness to outlier points. How would figures 5 and 6 look in this case? 6. Typos: a. Line 101: explicitely -> explicitly b. line 246: enveloppe -> envelope

Reviewer 3



# Update after rebuttal I thank the authors for their comments in the rebuttal. In conjunction with discussions with other reviewers, they helped me understand the salient points of the paper better. I would still add the following suggestions: 1. It would strengthen the paper if advantages over existing methods that are not Wasserstein-based could be showed (persistence landscapes, persistence images, ...), in particular since those methods have the disadvantage of suffering from descriptors that are not easily interpretable and do not yield 'proper' means (in the sense that the mean landscape, for example, cannot be easily converted back into a persistence diagram. 2. The experiment could be strengthened by discussing a more complex scenario that demonstrates the benefits of TDA _and_ the benefits of the proposed method. This could be a more complex data set, for example, that is not easily amenable to non-topological approaches, such as natural image data. I realize that this a balancing act, however. Having discussed at length how to weigh the originality of the paper, I would like to see the method published in the hopes that it will further promulgate TDA. I am thus raising my score. # Summary This paper presents an algorithm for calculating approximate distances between persistence diagrams, i.e. feature descriptors from topological data analysis. The approximation is based on a reformulation of the distance calculation as an optimal transport problem. The paper describes how to apply previous research, most notably entropic regularization, to make calculations efficient and parallelizable. In addition to showing improved performance for distance calculations, the paper also shows how to calculate barycentres in the space of persistence diagrams. Previously, this involved algorithm that are computationally infeasible at larger scales, but the new formulations permits faster calculations. This is demonstrated by deriving a $k$-means algorithm. # Review This is an interesting paper and I enjoyed reading it. I particularly appreciated the step-by-step 'instructions' for how to build the algorithm, which increases accessibility even though the topic is very complex with respect to terminology. However, I lean towards rejecting the paper for now: while the contribution is solid and reproducible, the paper appears to be a (clever!) application of previous works, in particular the ones by Altschuler et al., Benamou et al., Cuturi, as well as Cuturi and Doucet. In contrast to these publications, which have an extremely general scope (viz. the whole domain of optimal transport), this publication is only relevant within the realm of topological data analysis. Moreover, the experimental section is currently rather weak and does not give credence to the claim in the abstract that 'clustering is now demonstrated at scales never seen before in the literature'. While I agree that this method is capable of handling larger persistence diagrams, I think the comparison with other algorithms needs to have more details: the 'Hera' package, as far as I know, also permits the calculation of approximations to the Wasserstein distance and to the bottleneck distance between persistence diagrams. Were these used in the experiments? And if so, what were the parameters such as the 'relative error'? And for that matter, how many iterations of the new algorithm were required and what are the error bounds? In addition, it might be fairer to use a single-core implementation for the comparison here---this would also show the benefits of parallelization! Such plots could be shown in the supplementary materials of the paper. At present, the observations with respect to speed are somewhat 'obvious': it seems clear that a multi-core algorithm (on a P100!) can always beat a single-core implementation. The same goes for the barycentre calculations, which are not even written in the same programming language I assume. Instead of showing that the algorithm is faster here (which I will definitely believe) can the authors say more about the qualitative behaviour? Are the barycentres essentially the same? How long does convergence take? Statements of this sort would also strengthen reproducibility; the authors could show all required parameters in addition to their experiments. Minor comments: - L. 32: If different kernels for persistence diagrams are cited, the first one, i.e. the multi-scale kernel by Reininghaus et al. in "A Stable Multi-Scale Kernel for Topological Machine Learning" should also be cited. - In the introduction, works about approximations of persistence diagram distances (such as the one by Kerber et al.) could be briefly discussed as well. Currently, they only appear in the experimental section, which is a little bit late, I think. - L. 57: "verifying" --> "satisfying" (this happens at other places as well) - L. 114: $\mathcal{M}_{+}$ should be briefly defined/introduced before usage - How are parameters tuned? While I understand how the 'on-the-fly' error control is supposed to work, I am not sure how reasonable defaults can be obtained here. Or is the algorithm always iterated until the approximation error is sufficiently small? - L. 141: Please add a citation for the C-transform, such as Santambrogio (who is already cited in the bibliography) - L. 155: "criterion" --> "threshold" - L. 168: What does $|D_1|$ refer to? Some norm of the diagram points? This is important in order to understand the discretization error. - L. 178: From what I can tell, the algorithm presented in the paper generalizes to arbitrary powers; does the statement about the convolutions only hold for the squared Euclidean distance? The authors should clarify this. - L. 231: "Turner et al." is missing a citation - Timing information for the first experiment in Section 5 should be included; it is important to know the dependencies between the 'fidelity' of the approximation and the computational speed; in particular since the paper speaks about 'smearing' in L. 251, which prohibits the recovery of 'spiked' diagrams